# Non-Hemorrhagic Adrenal Infarction during Pregnancy: The Diagnostic Imaging Keys

**Pierre Chagué [1,2,\*]**, **Antoine Marchi [1]**, **Alix Fechner [1,2]**, **Ghina Hindawi [3]**, **Hadrien Tranchart [2,4]**,
**Julie Carrara [2,5]**, **Alexandre J. Vivanti [2,5]** and **Laurence Rocher [1,2,6]**

1   AP-HP, Hôpital Antoine Béclère, Service de Radiologie, 92140 Clamart, France; marchia@hotmail.fr (A.M.);
    alix.fechner@aphp.fr (A.F.); laurence.rocher@aphp.fr (L.R.)
2   Faculté de Médecine Paris-Sud, Université Paris-Saclay, 94270 Le Kremlin Bicêtre, France;
    hadrien.tranchart@aphp.fr (H.T.); julie.carrara@aphp.fr (J.C.); alexandre.vivanti@aphp.fr (A.J.V.)
3   AP-HP, Hôpital Kremlin Bicêtre, Service de Radiologie, 94270 Le Kremlin Bicêtre, France;
    ghina.hindawi@aphp.fr
4   AP-HP, Hôpital Antoine Béclère, Service de Chirurgie Digestive, 92140 Clamart, France
5   AP-HP, Hôpital Antoine Béclère, Service de Gynécologie-Obstétrique, 92140 Clamart, France
6   BIOMAPS, UMR 9011 Université Paris-Saclay, 4 pl. du Général Leclerc, CEDEX, 91401 Orsay, France
\*   Correspondence: pierre.chague@aphp.fr

**Abstract:** *Background:* non-hemorrhagic adrenal infarction (NHAI) is a rare cause of acute abdominal/flank pain during pregnancy; in order to ensure prompt and appropriate treatment, this diagnosis should not be overlooked. This case series highlights pertinent imaging findings, including ultrasounds (USs), computed tomography (CT), and magnetic resonance imaging (MRI) of recent NHAI cases. *Methods:* we compiled all consecutive NHAI cases from two university hospitals over a two-year period and checked the relevant clinical, laboratory, and imaging findings. Relevant articles on NHAI published from January 2010 to March 2021 were analyzed. *Results:* six cases were found in our database. CT-scans typically showed enlarged, hypodense, and non-enhanced adrenal glands. Unenhanced MRIs allowed for diagnoses and showed enlarged adrenal glands in the signal hyperintensity on T2 and diffusion-weighted imaging, without any signal hyperintensity on T1. In two of our six cases, USs showed swollen adrenal glands with fluid collection. *Conclusion:* NHAI and its differential diagnosis—in cases of acute pain during pregnancy—highlight the crucial roles of integrated radiological examination and cooperation between obstetricians and radiologists, both of whom should consider the location of the pain, the accessibility and tolerance of MRI, and the radiation exposure of CT. Despite its supposed poor sensitivity, an US performed because the patient reports pain should also be used to examine the adrenal gland regions. Non-enhanced MRI is clearly of value and access to it in emergencies is important to avoid radiation exposure

**Keywords:** adrenal infarction; abdominal pain in pregnancy; non-hemorrhagic adrenal infarction; adrenal thrombosis; adrenal ischemia; MRI; management

## 1. Introduction

Non-hemorrhagic adrenal infarction (NHAI) is a rare clinical cause of acute abdominal/flank pain during pregnancy. It should be differentiated from acute bilateral adrenal hemorrhage, which is often described in patients with hemostatic disorders. The clinical presentation of NHAI may vary from patient-to-patient, as well as the location of the pain (flank, abdomen, or chest). This could point to various common non-obstetric causes of acute pain, such as hydronephrosis, appendicitis, gallstones, and pulmonary embolism. Physicians and radiologists must agree on the need for imaging and define a challenging strategy. They should consider radiation exposure, diagnostic accuracy, and the availability of each imaging modality.

If sonography is inconclusive, computed tomography (CT) is often performed as a second-line test despite radiation exposure, but magnetic resonance imaging (MRI) seems

to be a good, non-irradiating alternative to establish the cause of pain [1]. Among several differential diagnoses, NHAI, a rare clinical entity, should not be overlooked, in order to ensure prompt treatment [2].

The purpose of this study, based on recent cases in two university hospitals, was to highlight the crucial role of imaging, including US, CT, and MRI. We also reviewed the previous cases reported in the literature.

## 2. Materials and Methods

### 2.1. Patients

Between January 2018 and December 2020, 282 pregnant women were referred to the radiological departments of two university maternity hospitals for exploration of acute flank/upper abdominal/back–chest pain. Both hospitals have radiological databases where all reports are entered. We searched for adrenal gland anomalies among these reports and retrieved/confirmed NHAI in six women from ages 19 to 38 years (mean 29.5 years). All clinical and laboratory data and follow-up findings were retrospectively collected. All data analyzed were collected as part of routine diagnosis and treatment. The institutional review boards approved this study, and because of the retrospective nature of the study design, they waived the requirements for informed patient consent.

### 2.2. Imaging Techniques

All patients had abdominal and pelvic ultrasound examinations as first-line imaging. They also had contrast-enhanced CT. In two patients, CT pulmonary angiograms were required to rule out pulmonary embolisms, followed by abdominal CTs, because of the pathological findings at the lower parts of the chest CTs. Abdominal CTs were performed in two other patients because they had a history of kidney stones; two patients underwent dual energy contrast-enhanced CTs because of suspected appendicitis. Five patients had abdominal 1.5 T MRIs without gadolinium injection (MAGNETOM Aera, Siemens, Germany).

### 2.3. Imaging Analysis

All images were interpreted and reviewed by two radiologists with 7 and 31 years of experience in urological imaging. The diagnoses were defined as enlarged and hypoenhanced adrenal glands on the CT scans. Non-hemorrhagic components were defined by the absence of hyperdensity on non-enhanced CT scans and the lack of increased signal intensity in T1-weighted MR imaging.

### 2.4. Literature Analysis

We searched the PubMed/MEDLINE databases for previous case reports using the keywords "non-hemorrhagic adrenal infarction", "adrenal thrombosis", and "pregnancy".

## 3. Results

### 3.1. Clinical and Laboratory Findings

We diagnosed NHAI in six women (mean age 29.5 years; range 19–38 years). In all cases, NHAI occurred during the third trimester of pregnancy (mean gestational age 32 weeks; range 26–37 weeks). None of the patients had a personal or family history of venous thromboembolism. Three had a history of kidney stones, including one during the first trimester. Pain was resistant with common analgesics, and opioid titration was required. Clinical and US pelvic examination findings were normal in all six patients and none were at risk of impending delivery. Laboratory tests showed leukocytosis (median $16.5 \times 10^9$/L; IQR 13.1–18.8) and increased C-reactive protein (median 50.5 mg/L; IQR 31–74.5). Urine analysis and culture were normal for all women. An extensive search for a hereditary or acquired thrombophilic disorder proved negative, except for patient 6, who had a bilateral adrenal infarct (positive lupus anticoagulant without anti-cardiolipin or anti-beta-2 glycoprotein antibodies). This included homocysteine, proteins C and S,

antithrombin III, factor II, VIII levels, lupus anticoagulant, anti-cardiolipin and anti-beta-2 glycoprotein antibodies, Leiden, prothrombin gene 20210A and JAK2 polymorphisms. Therapeutic dose anticoagulation was initiated for all patients. Patient 1 (37 weeks of gestation) had labor induction in order to manage analgesic treatments more easily. Patient 6 had a bilateral adrenal infarct; hydrocortisone supplementation started as soon as the diagnosis was confirmed by MRI. Under supplementation, plasma cortisol levels were near the lower limits of normal at 8 a.m. and 10 a.m. and the adrenocorticotropic hormone (ACTH) level was normal. She received an adapted treatment for the rest of her pregnancy and for delivery. Four months after her delivery, she still requires hydrocortisone supplementation. The other four patients had no further complications. Clinical, laboratory, and imaging findings are summarized in Table 1.

### 3.2. Imaging Findings

The initial abdominal and pelvic US showed pyelocaliceal dilatation and kidney stones in one patient and a swollen right adrenal gland in two others, including fluid collection in one of them. Typical US findings are shown in Figure 1 and Video S1.

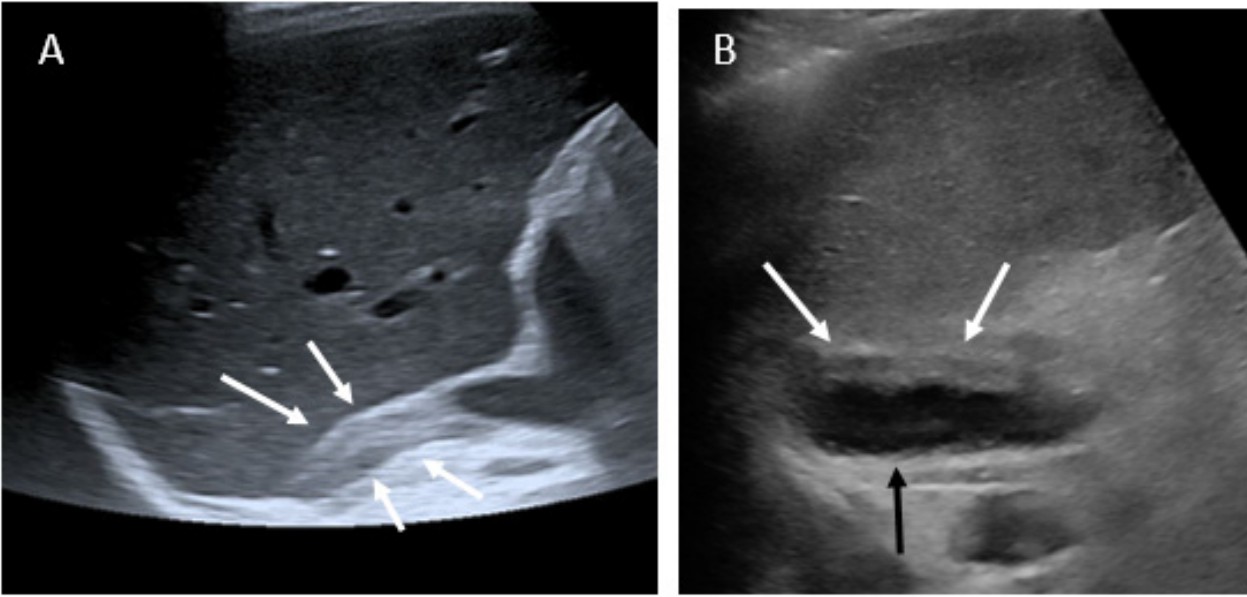

**Figure 1.** Ultrasound findings. (**A**): patient 1, enlarged right adrenal gland (white arrows); (**B**): patient 5, enlarged left adrenal gland (white arrows) associated with fluid collection (black arrow).

Video S1: patient 5 US findings; showing fluid collection beside the swollen left adrenal gland.

**Table 1.** Demographic data and findings.

| Patient | Age | Gravidity and Parity | Gest. Age | Clinical Presentation/Initial Suspected Diagnosis | Laboratory Findings | Imaging Findings | Side | Treatment | Imaging Follow-Up |
|---|---|---|---|---|---|---|---|---|---|
| 1 | 33y | G4P3 | 37w | Right-sided abdominal pain/appendicitis | Leukocytosis: 19x10*9/L, CRP: 49 mg/L, D-dimer: 1070 ng/mL | **US: swelling adrenal gland** MRI: typical findings ‡, without any diffusion imaging CT: typical findings † | Right | After giving birth, Oral anticoagulation and antiplatelet during 11 months | CT at 3 and 7 months: Atrophic adrenal with partially restored glandular enhancement |
| 2 | 38y | G3P1 | 26w | history of kidney stone/Right back flank pain/renal colic | Leukocytosis: 20x10*9/L, CRP: 17 mg/L | US: no abnormality CT: typical findings with vein thrombus | Right | Heparin and then oral anticoagulation during 6 months | MRI at 1 month and CT-enhanced at 3 months: Atrophic adrenal with partially restored glandular enhancement |
| 3 | 19y | G1P0/twin pregnancy | 32w | Previous left acute obstructive pyelonephritis during the same pregnancy/right back flank pain/renal colic/ | Leukocytosis: 18x10*9/L, CRP: 82 mg/L | US: Pyelocaliceal dilatation and kidney stones CT: typical findings with vein thrombus MRI: typical findings | Right | Heparin injection follow by oral anticoagulation for 3 months | CT at 3 months: isolated residual atrophy of the lateral arm of the gland |
| 4 | 34y | G1P0 | 31w | Right upper quadrant pain/hepatic colic or pulmonary embolism | Leukocytosis: 15x10*9/L, CRP: 25 mg/L, D-dimer: 1500 ng/L | US: No abnormality CT: typical findings. MRI: typical findings with fluid collection | Right | Heparin injection during the pregnancy | MRI at one week: no change. MRI at one month: Appearance of T1-weighted hyperintensity |
| 5 | 31y | G3P0 | 36w | Left upper back and low back chest pain/ pulmonary embolism | Leukocytosis: 12.4x10*9/L, CRP: 187 mg/L, D-dimer: 820 ng/L | **US: no initial abnormality/swelling adrenal gland and fluid collection on review** CT: typical findings. MRI: typical findings with fluid collection | Left | Heparin injection during the pregnancy | MRI at 4 months: swollen left adrenal gland. Collection decreased with partially restored glandular enhancement |

**Table 1.** *Cont.*

| Patient | Age | Gravidity and Parity | Gest. Age | Clinical Presentation/Initial Suspected Diagnosis | Laboratory Findings | Imaging Findings | Side | Treatment | Imaging Follow-Up |
|---|---|---|---|---|---|---|---|---|---|
| 6 | 22y | G1P0 | 30w | Left pain then one day later right flank pain/ appendicitis | Leukocytosis: 10.3x10*9/L, CRP: 52 mg/L | US: no abnormality CT: bilateral typical findings with right vein thrombus. MRI: bilateral typical findings | Right & Left | Heparin injection during the pregnancy | No follow-up (recent case) |

* Retrospective analysis of ultrasound views indicated unusual visualization of the right adrenal gland. † Typical CT findings: enlarged and hypoenhanced appearance of the adrenal gland. ‡ Typical MRI findings: restricted diffusion of the gland, increased T2 signal intensity with surrounding edema without T1 hyperintensity. Gest: gestational.

CT findings were typical in each case with an enlarged and hypoenhanced adrenal gland. There was also infiltration of the adjacent fat. In three cases, we identified the tail of the adrenal vein thrombus extending into the inferior vena cava. Typical CT findings are shown in Figure 2.

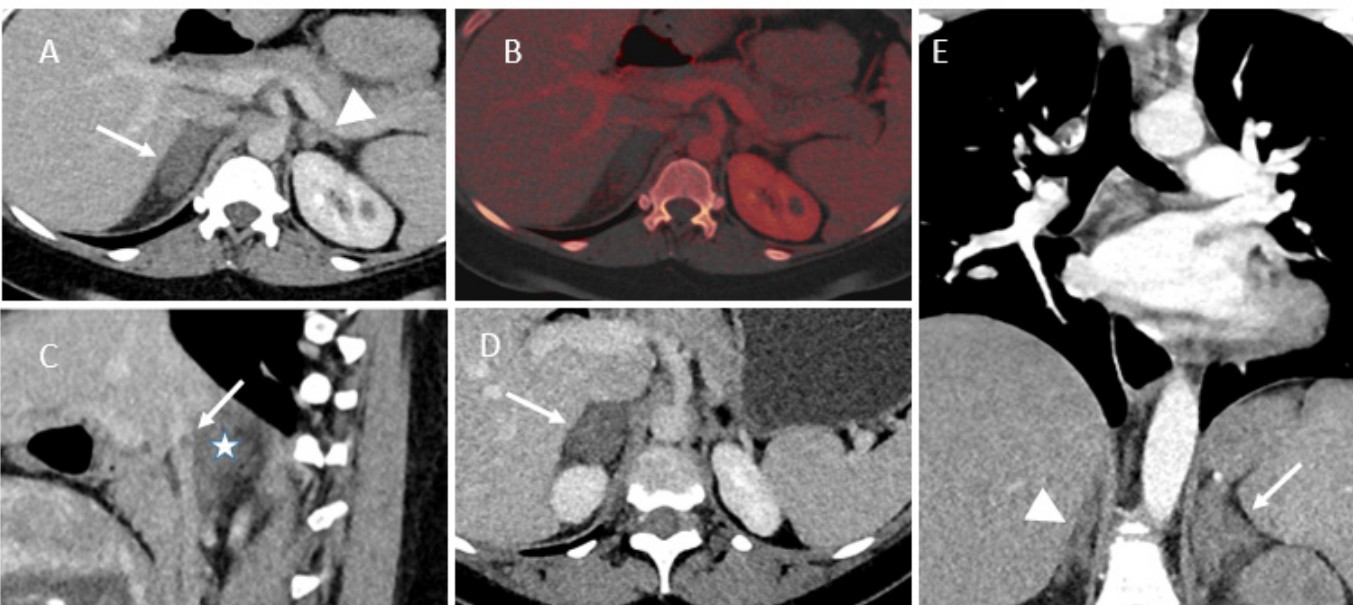

**Figure 2.** Contrast-enhanced CT findings of NHAI. (**A–C**) Patient 2. A: enhanced dual source CT showing an enlarged unenhanced right adrenal gland (arrow). Normal appearance of the contralateral adrenal gland (arrowhead). (**B**) Fusion of virtual non-contrast CT and iodine overlay image, demonstrating more easily the non-enhanced pattern. (**C**) Sagittal view of contrast-enhanced CT, showing the tail of the adrenal vein thrombus extending into the inferior vena cava (arrow) and the enlarged unenhanced right adrenal gland (star). (**D**) Patient 4, enlargement of the right adrenal gland. (**E**) Patient 5, CT pulmonary angiogram showing enlargement of the left adrenal gland and infiltration of the adjacent fat. Normal appearance of the contralateral adrenal gland (arrowhead).

Follow-up imaging most often showed atrophy of the ischemic adrenal gland. This atrophy may be partial. Glandular enhancement can also be partially restored (Figure 3). Patient 5, with initial fluid collection in the left adrenal gland, had a persistent swollen adrenal gland in the follow-up MRI at 4 months. The enhancement was partially restored, and the fluid collection had mostly shrunk in size.

For five patients, MRI, including diffusion and T1-weighted sequences, were performed to confirm diagnoses and the lack of an early hemorrhagic component of the adrenal infarction. In every case, the MRI showed edema of the adrenal gland and adjacent fat, no hyperintensity on T1-weighted imaging, and substantially restricted diffusion of the gland. Typical MRI findings are shown in Figures 4–6. Patient 6 had bilateral impairment. Clinically, she presented successive episodes of pain of the two flanks two days apart. The signal abnormalities may also be heterogeneous, especially on T2- and diffusion-weighted imaging (Figure 6).

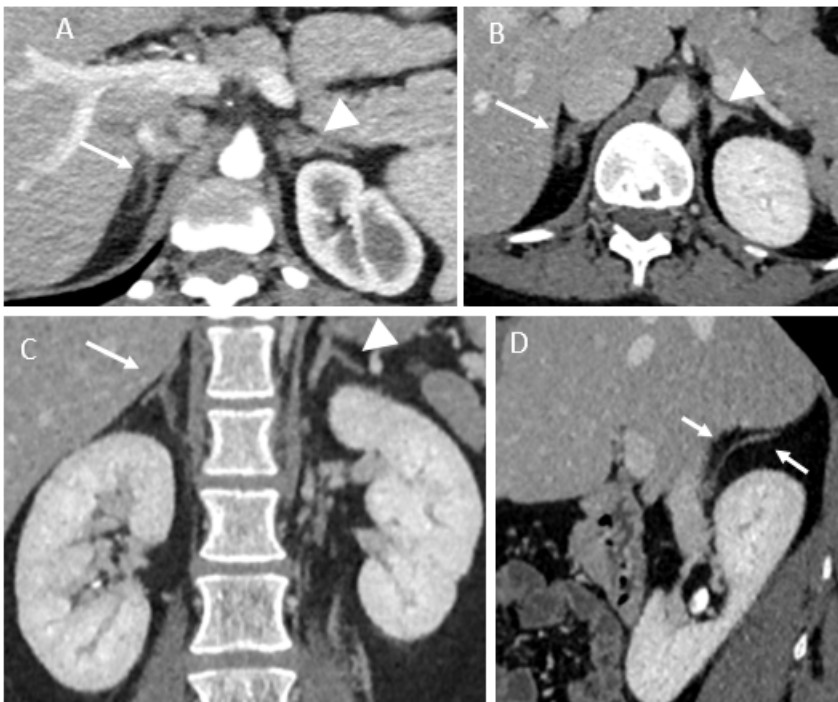

**Figure 3.** Follow-up imaging CT findings. (**A**) Patient 2, residual atrophy of the right adrenal gland (arrow). Normal appearance of the contralateral adrenal gland (arrowhead). (**B**) Patient 1, contrast-enhanced CT showing partial enhancement of the right adrenal gland (arrow). Normal appearance of the contralateral adrenal gland (arrowhead). (**C,D**) Patient 3, coronal and sagittal view of contrast-enhanced CT showing atrophy of the lateral arm of the right adrenal gland (arrow). Normal appearance of the contralateral adrenal gland (arrowhead).

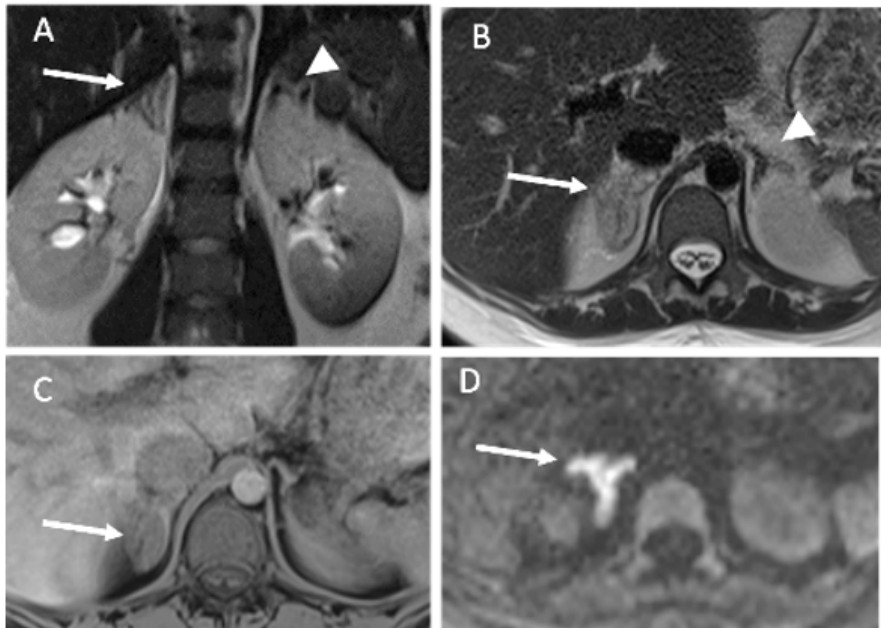

**Figure 4.** MRI findings of patient 3. (**A,B**) T2-weighted MR imaging, coronal (**A**) and axial (**B**) planes showing edema of the right adrenal gland (arrow) and adjacent fat. No abnormality of the contralateral gland (arrowhead). (**C**) Unenhanced axial T1-weighted imaging showing no hyperintensity of the adrenal gland (arrow). (**D**) Diffusion-weighted MR imaging (b800) showing restricted diffusion of the adrenal gland (arrow).

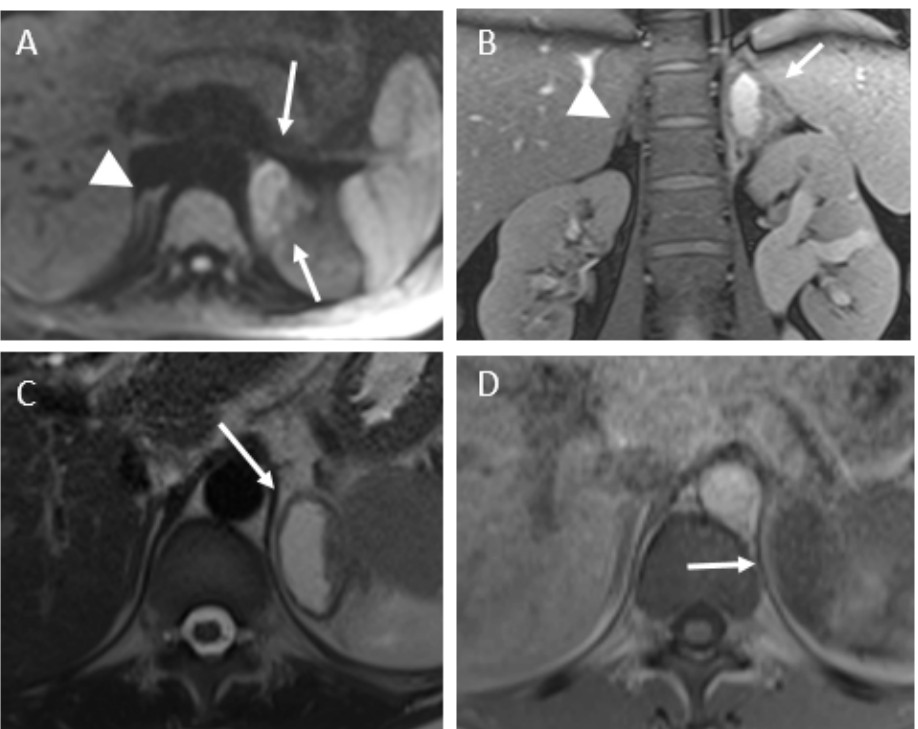

**Figure 5.** MRI findings of patient 3. (**A**) Diffusion-weighted MR imaging (b800) showing restricted diffusion of the adrenal gland (arrow). No abnormality of the contralateral gland (arrowhead). (**B**,**C**) Coronal and axial T2-weighted imaging, showing a fluid collection (arrow) beside the swollen left adrenal gland. *The left hydronephrosis is chronic (history of urinary stones).* (**D**) Axial unenhanced T1-weighted image, showing no hyperintensity of this fluid collection.

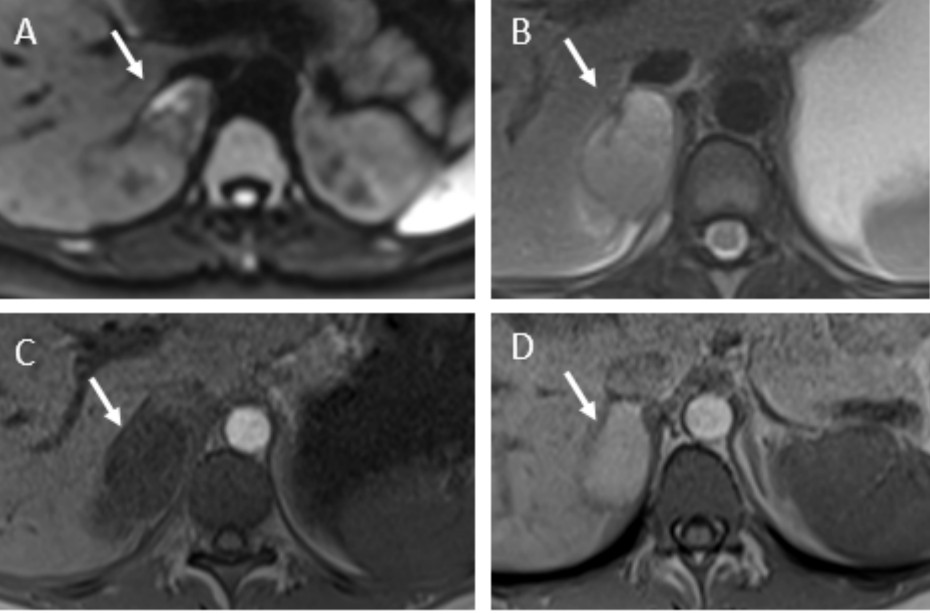

**Figure 6.** MRI findings of patient 4. (**A**) Diffusion-weighted MR imaging (b800) showing heterogeneous restricted diffusion of the adrenal gland (arrow). (**B**) Axial T2-weighted imaging, showing an enlarged right adrenal gland (arrow) with intermediate hyperintensity. (**C**) Axial unenhanced T1-weighted image, showing no hyperintensity of the adrenal gland (arrow). (**D**) Follow-up MRI at one month. Axial unenhanced T1-weighted image, showing appearance of hyperintensity of the adrenal gland (arrow) attributed to a secondary hemorrhage.

*3.3. Literature Review*

By March 2021, we had found 11 cases. Ten cases were recently summarized by Chasseloup et al. [3] and one more case of NHAI was described by Molière et al. [4]. NHAI is overwhelmingly found in pregnant women. Nevertheless, some cases are found without any context of pregnancy or in the immediate post-natal period [5].

In the literature, the clinical and laboratory presentations are quite similar from one patient to another. Among the 17 cases (including our 6 patients) that we compiled, 16 involved the right adrenal gland. Only one involved the left adrenal gland and two patients had sequential and bilateral infarction. These two patients have subsequently developed adrenal insufficiency secondary to the infarction [6]. CT was performed in at least 13/17 patients; MRI was performed in 9/17. The order in which the exams were performed is often not specified. Diffusion-weighted imaging findings have only been reported in two previous cases, showing a reduced apparent diffusion coefficient [4,7].

Pregnancy is a hypercoagulable state that predisposes the patient to thrombotic events. A pregnant woman has a relative risk of thrombosis five to six times higher than a non-pregnant woman of the same age [8]. Moreover, in four cases, we noted a hereditary or acquired thrombophilic disorder. Three patients were found to have increased factor VIII activity [3], one patient had a factor V Leiden mutation [9], and one patient was found to have a heterozygous C677T polymorphism in the gene encoding methylenetetrahydrofolate reductase (MTHFR) [10]. Three patients did not receive any therapeutic treatment. The others received anticoagulant therapy to prevent another sequential thrombosis. Among the seventeen patients described, eight had follow-up imaging at least one month after NHAI. Normalization of adrenal size was reported in four cases and persistent adrenal atrophy in four.

## 4. Discussion

The pathophysiology of NHAI remains unclear, but involves thrombosis of the adrenal vena. The histological analysis of adrenal infarction strongly suggest that the thrombosis of the main adrenal vein is the cause of the infarction. As *Fox* et al. suggested, the central vein thrombosis appeared well organized; thus, it seems more probable that the central vein thrombosis had resulted in sinusoidal thrombosis [11]. A possible explanation is that adrenal vein thrombosis can cause spasms of capsular arteries, resulting in cortical necrosis without hemorrhage. Adrenal glands are predisposed to micro-vascular thrombosis and infarction during procoagulable states. Elevated estradiol levels also increase the risk of thrombosis during pregnancy and contribute to the procoagulable state. The main involvement of the right adrenal gland, which was already highlighted in previous publications [3], appears to be due to blood stasis. This could be explained by the direct termination of the right adrenal vein in the inferior vena cava, in addition to a modification of the return of venous blood by the gravid uterus. Once diagnosed, patients should receive therapeutic anticoagulation, especially to prevent contralateral infarction [7]. The typical presentation of NHAI is acute onset of afebrile abdominal/flank pain. Although non-specific, the clinical presentation is similar from patient-to-patient—a sudden, severe, unremitting pain. The pain does not respond to regular pain medication used during pregnancy. Frequently, there is biological inflammatory syndrome. All six of our patients had elevated white blood cell counts and CRP. When NHAI is unilateral, there are no clinical or laboratory signs of adrenal insufficiency [3]. The location of the pain may vary from patient-to-patient, including back–chest, abdominal, and flank pain. This induces various supposed etiologies, including pulmonary embolism, renal/biliary colic, pyelonephritis, and appendicitis. Therefore, radiologists should be aware of this rare diagnosis and its variable clinical expression. Because of differential diagnoses, especially hydronephrosis, US screenings should remain as first-line imaging modalities, but in adults, it does not allow the diagnosis of NHAI with acceptable accuracy. Nevertheless, the unusual visualization of one adrenal gland should suggest its swelling, as in two of our cases. Because of its efficiency and safety, non-enhanced MRI should be the preferred second-line modality

for pregnant women [1]. It typically demonstrates restricted diffusion of the swollen gland and an increased T2 signal intensity, with surrounding edema without T1 hyperintensity. The signal abnormalities may be homogeneous or heterogeneous with multiple foci, especially on T2- and diffusion-weighted imaging [2]. Just as Molière et al. have shown with a control population, diffuse or multifocal high intensity on DWI is a landmark of adrenal ischemia [4]. Our imaging findings are consistent with previous descriptions in the literature. We noticed that two of our patients exhibited fluid collections beside the adrenal gland, and this pattern has never been described, to our knowledge.

MRI may help to differentiate physiological ureteral dilatation of pregnant women from pathological dilatation, with peripheral renal edema and renal enlargement. However, urinary stones can be challenging to identify on MRIs. CTs allow for specific diagnoses of renal colic due to ureteric stones and the functional consequences, which are characterized by a delayed nephrogram [12]. The latter sign cannot be assessed with the same accuracy by unenhanced MRI [13]. The use of contrast-enhanced CT should be limited during pregnancy, but it is nevertheless often performed due to lack of access to MRIs in emergency situations. Indeed, iodinated contrast medium is regularly used in pregnant women [14], especially to exclude pulmonary embolism. Gadolinium injection should be avoided in pregnant women, despite the absence of reported adverse effects. When there is a very strong indication for enhanced MRI, the smallest possible dose of one of the most stable gadolinium contrast agents should be used [14].

The main limitation of our study is the small sample of patients, despite the multicentric recruitment. This limitation is inherent to the rarity of the pathology. We are aware of the operator-dependent nature of ultrasonography, nevertheless, the adrenal area anomalies highlighted in our study were seen by two different sonographers. This should encourage practitioners to explore the adrenal aera, especially since modern ultrasound scans allow exploration of anatomic regions—areas that were previously supposed to be inaccessible.

For acute flank/abdominal pain during pregnancy, we suggest simplified imaging management. After an US, unenhanced abdominal and pelvic MRI should play an essential role in the etiological investigation of acute abdominal pain in pregnant women. If pulmonary embolism is initially suspected, the CT pulmonary angiogram remains essential, and radiologists should pay special attention to the adrenal glands, which are in the field of view. Abdominal/pelvic CTs should be performed if the MRI is inconclusive. Regarding our case series, abdominal CTs could have been avoided in every case if the MRI had been the second radiological exam after the US.

### 5. Conclusions

- Imaging plays a central role in the management of abdominal pain during pregnancy. The clinical presentation of NHAI is non-specific, but imaging findings are typical and highly similar from patient-to-patient.
- US examination performed for abdominal/flank pain should screen the adrenal gland region.
- Our results highlight the usefulness of unenhanced MRIs, which should include diffusion-weighted imaging.
- A prompt diagnosis of NHAI induced the implementation of anticoagulation and screening for acute adrenal insufficiency.

**Supplementary Materials:** The following are available online at https://www.mdpi.com/article/10.3390/tomography7040046/s1, Video S1: Ultrasound findings.

**Author Contributions:** Manuscript drafting: P.C. and L.R. Manuscript revision for important intellectual content: all authors. Approval of final version of submitted manuscript: all authors. Literature search: P.C., L.R. Study supervision: P.C., L.R. Acquisition of data: all authors. All authors have read and agreed to the published version of the manuscript.

**Funding:** This research received no external funding.

**Institutional Review Board Statement:** According to our institution legislation (APHP) and because of the retrospective analysis of the data, ethical approval was not required. The study was conducted according to the guidelines of the Declaration of Helsinki.

**Informed Consent Statement:** All patient hospitalized in the APHPs hospital are informed of the potential use of their data for research purpose. No patientin in our study refused the use of personal data. According to national (France) and institionnal (APHP) legislations, patient consent was waived due to the retrospective analysis of the data.

**Data Availability Statement:** The data presented in this study are available on request from the corresponding author. The data are not publicly available due to ethical and privacy.

**Acknowledgments:** The authors would like to thank the Department of Obstetrics and Gynecology, Bicêtre University Hospital Centre, AP–HP, which was involved in the management of two patients.

**Conflicts of Interest:** The authors declare that they have no competing interest.

**Ethical Statement:** The paper does not report on primary research. All data analyzed were collected as part of routine diagnosis and treatment. Therefore, according to our intuitional and national legislation, no explicit consent was needed.

## Abbreviations

| | |
|---|---|
| NHAI | non-hemorrhagic adrenal infarction |
| MRI | magnetic resonance imaging |
| IQR | interquartile range |
| CRP | C-reactive protein |
| FS | fat-suppressed |
| CT | computed tomography |
| GA | gestational age |
| ACTH | adrenocorticotropic hormone |
| US | ultrasound |

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
