# Peer review of "Non-Hemorrhagic Adrenal Infarction during Pregnancy: The Diagnostic Imaging Keys"

_tomography, doi:10.3390/tomography7040046_

Round 1

Reviewer 1 Report

I have found the paper very interesting. Enlarged hypodense suprarenal gland, not enhancing after c.m. and pain are rarelly related to hypovascular infarct. Restricted diffusion is the sign of early ischemia, but in suprarenal gland? Asymetry? I like the idea.

Author Response

Point 1: I have found the paper very interesting. Enlarged hypodense suprarenal gland, not enhancing after c.m. and pain are rarely related to hypovascular infarct. Restricted diffusion is the sign of early ischemia, but in suprarenal gland? Asymmetry? I like the idea.

Response 1: We sincerely thank you for your comment.

Indeed, the restricted diffusion is a sign of ischaemia. Molière et al. have compared adrenal ADC value in a control population to adrenal ADC value of NHAI and found a significant difference. The asymmetry is also subsequent.

We developed this idea in our discussion (L291-292)

Reviewer 2 Report

This is an intersecting paper presenting imaging findings including ultrasound (US), computed tomography (CT), and magnetic resonance imaging (MRI) of recent (non-hemorrhagic adrenal infarction (NHAI) in a small series of six patients.
The abstract represents the main manuscript. 
The manuscript is relatively well written.
The study is interesting. 
The conclusions sound balanced, and they correspond and enrich the previous publications in this field. 
I have trouble understanding if the paper is a review of literature or o a case series presentation. 
The presented information is of great value. I personally suggest rethinking the paper in the form of a pictorial essay. It will definitely have more value. 

Author Response

Point 1: This is an intersecting paper presenting imaging findings including ultrasound (US), computed tomography (CT), and magnetic resonance imaging (MRI) of recent (non-hemorrhagic adrenal infarction (NHAI) in a small series of six patients. I have trouble understanding if the paper is a review of literature or o a case series presentation. 
The presented information is of great value. I personally suggest rethinking the paper in the form of a pictorial essay. It will definitely have more value. 

Response 1: We sincerely thank you for your review and comment.

We are fully aware that our article could have been a pictorial essay. Nevertheless, several elements made us prefer the form of the original article.

Contrary to a pictorial essay, we chose to study a single pathology (NHAI) on a homogeneous target population (pregnant women with acute pain). We did not want to make general descriptions on the aetiologies of acute pain nor on the adrenal pathology. The fact that the pathology is rare, and our inclusion period limited, makes our sample size small (this is a limitation of the study). We have added the limitations to our discussion (Lines 309-314).

We bring new elements in the description of this pathology, especially in ultrasound, which is more like the original article.

It is not a review either because we bring new cases to the pathology and new elements.

The two other reviewers having found the format adapted we are embarrassed and cannot easily modify it.

Nevertheless, thanks to your remarks we have brought more limitations to our study in the discussion section and enriched it (Lines 260-268, ..., 308-314)

Reviewer 3 Report

Dear Authors:

The authors have carried out a case series study  of all consecutive non-hemorrhagic adrenal infarction (NHAI) during pregnancy from two university hospitals over a two-year period and checked the relevant clinical, laboratory and imaging findings. The  aim of this study was to evaluate the crucial role of imaging, including  ultrasound,  computed tomography and  magnetic resonance imaging and conduct a literature review of the previous cases reported.

Some considerations need to be taken into account:

  • This manuscript is a fine, interesting and well designed study with an important limitation due to its small sample size (6 patients included over a two-year period) which reduces the interest of the study to a certain extent.
  • The abdominal and pelvic ultrasound examination was performed by the same specialized radiologist? The results of the ultrasound are radiologist dependent and the generalization of this procedure could imply a selection bias and diminish the power of the results
  • Nice video of ultrasound findings although It is executed very fast (line 117)
  • The pathophysiology of NHAI should be analyzed in more depth in the text since it is only mentioned in a very brief way at the beginning of the discussion. Because of the paucity of cases in literature, the precise pathogenesis of  thrombosis during pregnancy is largely unknown and requieres a broader discussion.
  • Blood stasis can be attributable to  blood flow stasis secondary to inferior vena cava compression by the gravid uterus. The greater involvement of the right adrenal gland could be due to a direct drainage of the right adrenal vein into the vena cava and  length of the adrenal vein. Have you found similar data in the literature review carried out?
  • The procoagulant state of pregnancy is not mentioned in the discusión. Only blood flow stasis secondary to inferior vena cava compression by the enlarged uterus is  briefly discussed, but physiologic changes that affect coagulation factors have been omitted and could play a relevant role in the genesis of this process. It is only mentioned briefly in the literature review (line 202).

In conclusion, it is a well designed study that needs revisión.

Kind regards

Author Response

Point 1: This manuscript is a fine, interesting, and well-designed study with an important limitation due to its small sample size (6 patients included over a two-year period) which reduces the interest of the study to a certain extent.

Response 1: We sincerely thank you for your review and comment. Indeed, our sample size is small. The pathology is rare, our inclusion period limited, and our homogeneous population of pregnant women make difficult the recruitment of a larger sample. This a limitation of the study, that we have added to our discussion (L308-309)

Point 2: The abdominal and pelvic ultrasound examination was performed by the same specialized radiologist? The results of the ultrasound are radiologist dependent, and the generalization of this procedure could imply a selection bias and diminish the power of the results

Response 2: Thank you for pointing out this. No, the ultrasounds were not performed by the same radiologist. Indeed, our study is multicentric and the recruitment was done retrospectively. We have added in our discussion the operator-dependent nature of the ultrasound. However, we are encouraged that the ultrasound abnormalities were seen by two different radiologists, which increases the external validity of the study (Lines 308-314).

Point 3: Nice video of ultrasound findings although it is executed very fast (line 117)

Response 3: We have slowed down the video scrolling speed

Point 4: The pathophysiology of NHAI should be analysed in more depth in the text since it is only mentioned in a very brief way at the beginning of the discussion. Because of the paucity of cases in literature, the precise pathogenesis of thrombosis during pregnancy is largely unknown and requires a broader discussion.

Response 4: As suggested by the reviewer, we have now developed more deeply the physiopathology in our discussion. (Lines 260-268)

Point 5: The greater involvement of the right adrenal gland could be due to a direct drainage of the right adrenal vein into the vena cava and length of the adrenal vein. Have you found similar data in the literature review carried out?

Response 5: Indeed, as we noted in the discussion, this predominance of right-sided infarctions during pregnancy has already been discussed. (lines 270-273).

Point 6: The procoagulant state of pregnancy is not mentioned in the discussion. Only blood flow stasis secondary to inferior vena cava compression by the enlarged uterus is briefly discussed, but physiologic changes that affect coagulation factors have been omitted and could play a relevant role in the genesis of this process

Response 6: Thanks to you we have developed this point in our discussion (L266-268)

Round 2

Reviewer 2 Report

I appreciate the authors' efforts to address the limitations.  The paper is well written.  I am in favor of publication if the Editor agrees.